# Significant Sex Differences in the Efficacy of the CSF1R Inhibitor-PLX5622 on Rat Brain Microglia Elimination

**DOI:** 10.3390/ph15050569

**Published:** 2022-05-02

**Authors:** Aviv Sharon, Hadas Erez, Micha E. Spira

**Affiliations:** 1Department of Neurobiology, Alexander Silberman Institute of Life Science, The Hebrew University of Jerusalem, Jerusalem 9190401, Israel; aviv.sharon1@mail.huji.ac.il (A.S.); hadas.erez@mail.huji.ac.il (H.E.); 2The Charles E. Smith Family and Joel Elkes Laboratory for Collaborative Research in Psychobiology, The Hebrew University of Jerusalem, Jerusalem 9190401, Israel

**Keywords:** microglia, CSF1R, PLX5622, sex differences, immunohistology, rat

## Abstract

Microglia play pivotal roles in central nervous system development, homeostasis, responses to trauma, and neurodegenerative and neuropsychiatric disorders with significant sex-bias in their symptoms and prevalence. Survival of the microglia in adult brains depends on the expression of the colony-stimulating factor 1 receptor (CSF1R). The inhibition of CSF1R by brain-permeant PLX5622 in the chow eliminates, within 5–10 days, ~90% of the microglia in female and male mice, thereby enabling the investigation of the roles of the microglia in health and pathological mice models. Because of a prevailing “impression” that PLX5622 is ineffective in rats, it has hardly been used in studies of adult rats. Here, we report that effective microglia elimination by PLX5622-chow in rats is highly sex-dependent. Our observations provide missing information for the limited use and interpretation of PLX5622 in biomedical studies of the microglia in rat models. The sex differences that are too often overlooked must be carefully considered and clearly emphasized.

## 1. Introduction

Microglia, the resident immune cells of adult mammalian brains, play pivotal roles in central nervous system development, homeostasis, brain responses to infections or sterile insults, and neurodegenerative and neuropsychiatric disorders with significant sex-bias in their symptoms, onset, and prevalence [1,2,3,4,5,6,7,8]. Under steady-state conditions, the microglia morphology in adult brains is characterized by branching filopodia that dynamically “surveil” the microenvironment. Although this morphology and function is common to all “resting microglia”, specific cell geometries can be identified in various brain regions [9]. Mice microglia are characterized by months-long total turnover rates [10,11,12], with the microglia population density maintained by local proliferation [12]. The steady-state dynamics, morphology, and transcriptional phenotypes of the microglia are rapidly altered in response to different types of brain pathologies, trauma, and infections [13].

The discovery that the survival of the microglia in adult brains depends on the expression of the colony-stimulating factor 1 receptor (CSF1R), along with the development of toxins, pharmacology, and gene-based approaches to blocking or eliminating the expression of CSF1R, have all facilitated the exploration of the microglia’s roles in health and disease [14,15,16]. The recent development of effective blood–brain-barrier (BBB)-permeant CSF1R inhibitors such as PLX3397 (Pexidartinib) and PLX647 has significantly advanced this field [15,16,17,18]. Further momentum has derived from the development of PLX5622, which exhibits higher specificity to CSF1R and improved BBB penetrance (20%) than earlier generations of brain-permeant CSF1R inhibitors. The delivery of PLX5622 in rodent chow (1200 mg/Kg) depleted the microglia in the mouse cortex, hippocampus, and thalamus by 80–90% within 5–7 days [19]. The use of PLX3397 in mouse chow confirmed the microglial loss as genuine and not representing the downregulation of microglial markers. The microglia population recovers on withdrawal of the PLX3397 diet [17,18]. The depletion was due to cell death and was dose- and time-dependent [17,19,20,21,22]. Immunohistological and Western blot analyses revealed that the elimination of the microglia by CFSR1 inhibitors is not associated with changes in the number of oligodendrocytes, neurons, and astrocytes and that mice depleted of the microglia show no behavioral or cognitive abnormalities [17,18,23].

PLX5622 chow has been extensively and effectively used in mice and nonhuman-primates (over 150 studies have been published since 2015). In spite of the indispensable use of rats in various aspects of brain research [24,25], PLX5622 chow has hardly been used to examine the roles of the microglia in adult rats (to the best of our knowledge, only four publications used PLX5622 in rats models). This odd fact is attributed to the “impression” that PLX5622 is “ineffective in rats” [26]. Examination of these four studies raised critical questions as to the possibility that the effectiveness of PLX5622 on the microglia elimination of adult female and male rats differs.

Briefly, the study by Spangenberg et al. [19] reported pharmacokinetic differences among mice and rats and among female and male rats in response to a single intravenous injection or one oral gavage session. This pharmacokinetic study, nonetheless, did not examine the outcome of treating adult male and female rats by PLX5622 for a number of days on microglia densities. In contrast to plausible pharmacokinetic predictions based on Spangenberg’s (2019) study, Riquier and Sollars [27] reported that the intraperitoneal injection of PLX5622 to both adult male and female rats depleted the microglia density from the gustatory system by 80–90% within 3 days, by 93% after 7, and to >96% after 14 days of IP injections. Importantly, no sex differences were reported. In an apparent inconsistency with these findings Liu et al. [28], reported that in male rats, 5 days of ad libitum PLX5622-chow led to a small ~40% reduction in Iba-1 immunoblots of the microglia in the spinal cord, while parallel experiments on mice by the same authors showed that all the microglia were eliminated [28]. Adding to the apparent contradictions are observations from our laboratory [29] that feeding female rats ad libitum with PLX5622-chow led within 5 days to a 79% elimination of cortical microglia and then further to ~94% and ~95% eliminations between days 12 and 21, respectively. Taken together, the fragmented information described above cannot serve as a solid background for designing studies in which PLX5622 is used in rats to decipher biomedical- and sex-related aspects of microglia function.

In view of the roles of the microglia in normal and pathological brains, in the apparent sex-bias of the microglia in neuro-disorders, the high efficacy of PLX5622-chow in eliminating the microglia, and the indispensable use of rats as a model system for brain research, we carried out the characterization of the effects of PLX5622 chow on the elimination of the microglia in female and male rats. The results presented here provide essential information for the use, limitations, and interpretation of PLX5622 in biomedical studies of the microglia using female and male rats.

## 2. Results

### 2.1. Steady-State Densities and Morphological Phenotypes of Microglia in the Brains of Female and Male Rat

Microglia densities and morphological phenotypes were mapped in adult female and male Sprague Dawley rats using Iba-1 antibody immunohistological labeling (Figure 1). The immunohistological identification of the microglia was confirmed by co-labeling Iba-1-positive cells with rat TMEM119 antibody that recognizes microglia-specific transmembrane proteins (e.g., [29,30]). Microglia densities in the cortex, hippocampus, amygdala, striatum, cerebellum, and olfactory bulb were mapped using stacks of confocal microscope images and the HCA-Vision/Acapella software.

As reported for mice brains [31], differences in microglia densities were recorded in different regions in the rat brain (Figure 2, Appendix A). In addition, some of these regions showed small but significant sex differences (*p* = 1.28 × 10^−6^, 5.1 × 10^−4^, and 0.007 in the cortex, amygdala, and striatum, respectively, Figure 2, Appendix A, [4,32]). As in mice, in both female and male rats, the lowest microglia densities were found in the cerebellum (235 ± 17 cells/mm^2^ in males and 228 ± 25 mm^2^ in females) and the highest in the olfactory bulb (391 ± 36 cells/mm^2^ in males and 381 ± 34 mm^2^ in females).

Earlier studies have documented that, at the steady-state, the microglia in all brain regions extend delicate ramifying branches (Figure 1 and Figure 3). Yet, quantitative morphological analysis of the mouse microglia has revealed characteristic morphological differences in different brain regions [31,33,34].

Our analysis of the microglia, shown in Figure 3 and Figure 4, confirmed morphological differences in different brain regions and among male and female rats. The average number of branches extending from a single cell body (roots/cell) was similar in males and females in all examined regions but was significantly smaller in the cerebellum and olfactory bulb (*p* ≤ 2.51 × 10^−18^ for cerebellum compared to other areas and *p* ≤ 2.49 × 10^−6^ for olfactory bulb compared to other areas—Appendix A). The averaged total microglia branch length, and the number of segments, branching points, and extremities were similar among males and females in the cortices, striata and olfactory bulbs (Figure 4, Appendix A, Appendix A), but there were sex differences in other brain regions (*p* ≤ 2.5 × 10^−8^ for cerebellum, *p* ≤ 1.39 × 10^−7^ for hippocampus, and *p* ≤ 0.0051 for amygdala for branch length and branching points, Appendix A).

### 2.2. Sex-Related Microglia Elimination and Morphological Alterations of the Surviving Microglia

Analysis of the effects of PLX5622 on the microglia was conducted on female and male rats fed with PLX5622 chow for 10 days. This point in time was selected as, in an earlier study (Sharon et al., 2021), we established that the maximal microglia elimination was achieved in female rats between 7 and 12 days of PLX5622 feeding.

Our study showed that the microglia densities in both female and male rats were generally reduced following 10 days of ad libitum feeding with PLX5622 chow (1200 mg/kg). Unexpectedly, however, the elimination of the microglia population in females was significantly larger (*p* ≤ 8.11 × 10^−19^) than in males (*p* ≤ 6.21 × 10^−9^, excluding the olfactory bulb, Figure 2 and Appendix A). In **female** cortices, hippocampi, amygdalas, and cerebella, the microglia density was reduced to 0.1–0.16 of the control values (Figure 2, Appendix A), to 0.34 in the striatum, and to 0.52 in the olfactory bulb (Figure 1 and Figure 2). In **males**, on the other hand, the microglia density was significantly less reduced only to 0.65–0.76 of the control value in the cortices, hippocampi, amygdalas, striata, and cerebella and was unchanged in the olfactory bulb (Figure 1 and Figure 2, Appendix A).

As the elimination of the microglia by CSF1R inhibitors is known to be dose-dependent, and as the effect of PLX5622 chow on males and females rats differed, we examined whether the microglia surviving after 10 days of PLX5622 feeding in both sexes had undergone any structural changes and, if so, whether the changes were similar. Morphological analysis of these microglia revealed that, except for the number of roots/cell, all other parameters (number of branches, segments, extremities, and branch lengths) were reduced in relation to the control in both females and males (Figure 4, Appendix A, Appendix A). The morphological changes were more pronounced in females than in males, but the averaged differences among males and females were small.

## 3. Discussion

In view of the widespread use of the CSF1R inhibitors PLX5622 in the form of rodent chow in studying the roles of the microglia in health and pathological conditions [1,14,16,17,19,35], and because of uncertainties (detailed in the introduction) as to the effects of PLX5622 on the adult rat microglia [19,26,27,29], we examined here two related questions: (a) are there any differences in the density distribution of microglia in adult female and male rat brains, and (b) are there sex differences in the response of adult rat brains to PLX5622 chow treatment? These questions are important, as rats are preferable to mice models for in vivo electrophysiological and behavioral studies, mainly because of their body size.

Our main observations were that: at the steady-state, the average microglia density is similar in female and male rat brains (although the small differences observed in the cortex, amygdala, and striatum were statistically significant; *p* ≤ 0.01, Appendix A). Microglia elimination in adult female rats proceeded within 10 days to 0.1–0.16 of the control levels in the cortex, hippocampus, amygdala, and cerebellum, in the striatum to 0.34, and only to 0.52 in the olfactory bulb. In contrast, 10 days of ad libitum PLX5622-chow feeding of male rats did not lead to a reduction in the average microglia density in the olfactory bulb (0.96), and there was significantly less (*p* ≤ 8.11 × 10^−19^ for females and *p* ≤ 6.21 × 10^−9^ for males, excluding the olfactory bulb) reduction than in females (0.65–0.76) in other brain regions (Figure 1 and Figure 2, Appendix A).

The sex-related differences in the effects of PLX5622-chow on microglia densities could reflect pharmacokinetic differences among females and males, as suggested by a single intravenous injection or single oral gavage of PLX5622 by Spangenberg et al. [19]. As the microglia densities in the male rat brain (apart from the olfactory bulb) were reduced by PLX5622 feeding (Figure 1 and Figure 2), and as the structural phenotypes of the surviving microglia were altered similarly to those in females (Figure 3 and Figure 4 and Appendix A), it is conceivable to assume that PLX5622 provided in the chow did cross the male rat BBB to reach the brain parenchyma. As the effect of CSF1R inhibition on microglia survival is dose-dependent [20], the observed sex differences in the number of eliminated microglia may reflect differences in the effective PLX5622 concentration in the brain parenchyma of females and males fed ad libitum. Interestingly, however, Riquier and Sollars [27] explicitly stated that no gender differences were noticed in microglia elimination in the rat gustatory system following intraperitoneal injections of PLX5622 in adult rats. It is thus conceivable that the introduction of PLX5622 by the intraperitoneal injection protocol rather than by feeding bypass pharmacokinetic barriers to elevate the effective PLX5622 brain concentration to a level that blocked the CSF1R in both males and females. Nonetheless, as differences in microglia elimination levels in some brain regions were significantly larger than in others (e.g., in male olfactory bulb, microglia were not eliminated, while, in other regions, the population was reduced to 0.65–0.76 of the control, and in females, the microglia elimination level in the olfactory bulb was smaller than in other regions, Figure 2), we cannot rule out the possibility that some differences in CSF1R expression levels, receptors’ affinities to PLX5622, or differences in downstream molecular cascades of CSF1R activation in male and female rats account for the gender differences. Currently, we have not attempted to differentiate among the above mechanisms. It should be noted that as common CSF1R mutations can bind CSF1, but cannot support kinase activity [35], the three proposed mechanisms could equally account for the documented gender differences.

From a practical point of view, we conclude that the effectiveness of the PLX5622-chow on the microglia elimination of adult females and male rats significantly differ. These differences should be taken into account and emphasized when planning and interpreting results conducted using PLX5622 on adult female and male rats.

## 4. Materials and Methods

### 4.1. Animals

Here, 12-week-old male and female Sprague Dawley rats (240–340 g) from the same litter were used for all experiments. All procedures were approved by the Committee for Animal Experimentation at the Institute of Life Sciences of the Hebrew University of Jerusalem. For microglial ablation, the rats were fed ad libitum, for 10 days, with a PLX5622 diet (1200 PPM PLX5622, Plexxikon Inc., Berkeley, CA, USA). PLX5622 was provided by Plexxikon Inc. and formulated in AIN-76A standard chow by Research Diets Inc.

### 4.2. Tissue Processing for Immunohistology

Control rats and rats fed ad libitum with PLX5622-chow for ten days were sacrificed for immunohistological examinations of the microglia as previously described by us [29]. When the rats stopped breathing, they were transcardially perfused with phosphate-buffered saline (PBS) at a rate of 10 mL/min for 40 min. This was followed by perfusion with 4% paraformaldehyde in PBS (PFA, Sigma-Aldrich) at a rate of 10 mL/min for 40 min. Next, the skulls were removed and the brain was post-fixed at 4 °C for an additional 12–24 h in 4% PFA. Afterward, brains were washed in PBS and incubated for 1–3 days in a 30% sucrose solution in PBS at 4 °C.

### 4.3. Cryosectioning and Immunohistological Labeling

To prepare for cryosectioning, cubes of the fixed tissues were isolated from different brain regions. These were placed in a freezing medium (Tissue-Plus O.C.T. Compound, Scigen) and frozen at −80 °C. The frozen tissue was then sectioned at 40 μm in the coronal plane using a Leica CM1850 Cryostat. Individual slices were collected and placed in 24-well plates containing PBS and processed as described earlier by us [29,30].

Microglia were labeled using rabbit anti-Iba-1 monoclonal antibody (Abcam ab178846, 1:2000). To confirm that the Iba-1 antibody labeled resident microglia and not infiltrated macrophages, we co-labeled the Iba-1-positive cells using goat anti-Iba-1 antibody (Abcam ab5076, 1:125) with rabbit polyclonal antibody for rat TMEM119 that recognizes microglia-specific transmembrane proteins (Synaptic Systems GmbH, 1:1000, 400 203). For additional details, see [29,30].

### 4.4. Microscopy

Confocal image stacks of the immunolabeled slices were acquired with an Olympus FLUOVIEW FV3000 confocal scan head coupled to an IX83 inverted microscope, using a 20× air objective (NA = 0.75). Sections were scanned in sequential mode.

Image stacks were acquired from 6 different brain areas: cortex, hippocampus, amygdala, striatum, cerebellum, and olfactory bulb. For microglia morphology analysis, we set a ×2 zoom.

### 4.5. Image Processing, Analysis, and Statistics

The image processing was implemented using the Fiji distribution of ImageJ [36,37], as follows. A maximum intensity projection image was created using 10 consecutive optical sections from each of the 40 μm thick brain slices. The cells were counted manually from each image. In total, brains of 6 female and 6 male rats fed with PLX5622 chow and corresponding controls were examined. Four hemispheres from each group were quantitatively analyzed in detail. The sample size of the immunohistological sections is given in Appendix A. Significant statistical differences between the different treatments (females and males, PLX5622, and control diet) were determined by a *t*-test for two samples assuming unequal variances. For all tests, a *p* value <0.01 indicated a statistically significant difference (Appendix A).

### 4.6. Microglia’s Morphology Analysis

A maximum intensity projection image was created using the Fiji distribution of ImageJ [36,37]. All consecutive optical sections from each of the 40 μm thick brain slices were used (from stacks of ×2 zoom). The file was saved as an 8-bit gray image (e.g., see Appendix A).

Those images were imported to the HCA-vision software (CSIRO Australia, [38,39]). The microglia cell bodies and extensions were automatically identified and measured (Appendix A). The parameters chosen for the analysis of the microglia morphology were: (a) Number of roots—the number of points where an extension structure touched a microglia cell body. (b) Number of branch points—the number of points where a microglia extension structure split into 2 or more branches. (c) Number of segments—a segment is a linear structure between branching points or a microglia cell body. (d) Number of extremities—the number of terminating extension segments. (e) Total extension length—sum of the length of all extensions. The images shown in Figure 1 and Figure 3 were acquired from different brain areas of the same animal.

For the analysis, we chose only cells that were properly recognized by the software and whose extensions were not cut at the edges of the image. For the automated microglia morphology analysis, we used the same optimal parameters setting for each group (male, female, control, and PLX5622-fed rats). The quality of the automated detection by these parameters setting is presented in Appendix A. The final data and group sample size are given in Appendix A.

## 5. Conclusions

We conclude that the effectiveness of PLX5622-chow on the microglia elimination of adult females and male rats significantly differ. These differences should be taken into account and emphasized when planning and interpreting results conducted using this and possibly other CSF1R antagonists on adult female and male rats.

## Figures and Tables

**Figure 1 pharmaceuticals-15-00569-f001:**
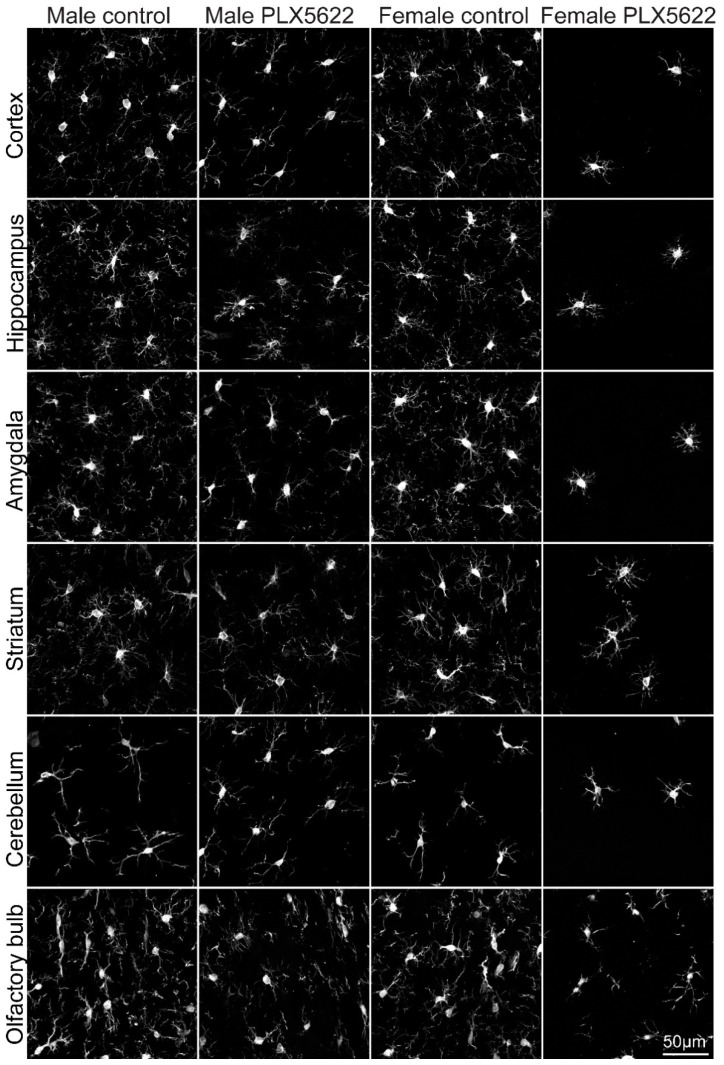
Confocal microscope images of Iba-1-labeled microglia in different brain regions in control male and female rats and rats fed ad libitum for 10 days on PLX5622 chow. The images shown were acquired from different brain areas of the same animal.

**Figure 2 pharmaceuticals-15-00569-f002:**
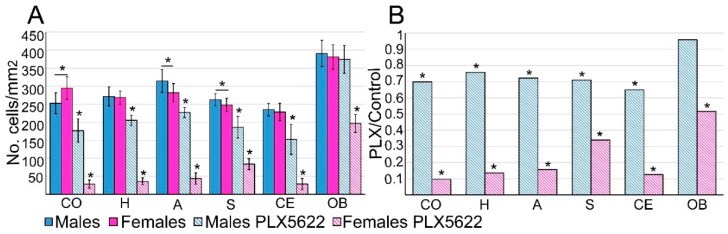
Comparison of microglia densities in control male and females and following 10 days of ad libitum PLX5622 feeding. (**A**) The average microglia densities (number of cells/mm^2^) in different brains regions in control male and female rats, and rats fed for ten days ad libitum on PLX5622 chow. Controls (homogeneous columns, male in blue and female in magenta) showed small but significant (*p*
**=** 1.28 × 10^−6^, 5.1 × 10^−4^ and 0.007 in the cortex, amygdala, and striatum, respectively) density differences between females and males within given brain regions. Large and significant differences (*p* ≤ 8.11 × 10^−19^ for females and *p* ≤ 6.21 × 10^−9^ for males, excluding the olfactory bulb) in the elimination of microglia from males and females were observed in all brain regions in response to PLX5622 chow (in diagonal stripes, Appendix A). Note that in the male olfactory bulb, the density of microglia was not altered by PLX5622 (*p* = 0.09). (**B**) Ratio values of averaged microglia densities in PLX5622-treated rats and control rats in different brain regions showed a significantly smaller effect of PLX5622 chow on male microglia densities than in females. CO—cortex, H—hippocampus, A—amygdala, S—striatum, CE—cerebellum, OB—olfactory bulb. Data are presented as mean number of cells/mm^2^ ± one standard deviation. ≥18 slices were prepared from the different brain regions of the different hemispheres (4 controls and 4 PLX5622-fed rats) of each sex. Significant differences were determined by a *t*-test for two samples assuming unequal variances. *p* value < 0.01 indicated a statistically significant difference. Underlined asterisks indicate statistical significance between males and females; asterisks indicate statistical significance between control and PLX5622-fed rats; vertical lines correspond to ± one standard deviation.

**Figure 3 pharmaceuticals-15-00569-f003:**
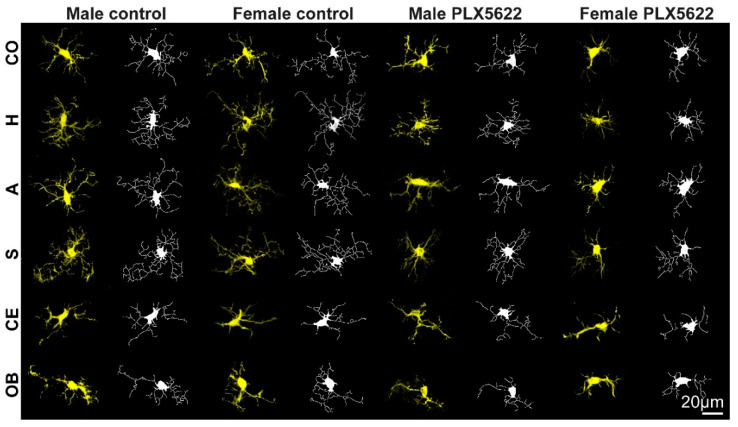
Maximum intensity projection confocal microscope images (yellow) from different brain regions in control male and female and in rats fed ad libitum for 10 days on PLX5622 chow. These were imported to the HCA-vision software (white) and the microglia cell bodies and extensions were automatically identified and measured. CO—cortex, H—hippocampus, A—amygdala, S—striatum, CE—cerebellum, OB—olfactory bulb (for statistics, see Figure 4 and Appendix A).

**Figure 4 pharmaceuticals-15-00569-f004:**
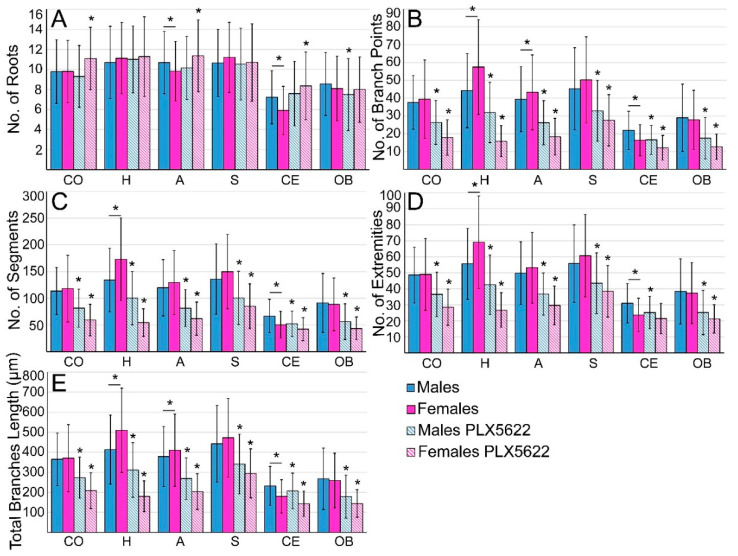
Quantitative morphological analysis of microglia in different brain regions in control male and female rats and rats fed ad libitum for ten days on PLX5622 chow. Controls (homogeneous columns, male in blue and female in magenta), PLX5622-treated rats (in diagonal stripes). (**A**) The number of roots/cell, (**B**) number of branching points, (**C**) number of segments, (**D**) number of extremities, and (**E**) total branch lengths. The number of roots/microglia in control and microglia-surviving PLX5622 in the cerebellum and olfactory bulb was lower than in other brain regions and was similar in males and females in most of the brain regions that were examined and was not altered by PLX5622 treatment. Microglia surviving the 10 days of PLX5622 chow showed a reduced number of branching points, segments, extremities, and total branch lengths in all examined brain regions. CO—cortex, H—hippocampus, A—amygdala, S—striatum, CE—cerebellum, OB—olfactory bulb. Data are presented as mean ± one standard deviation. ≥138 cells were analyzed from slices prepared from the six different areas of interest of the different hemispheres (4 controls and 4 PLX5622-fed rats) of each sex. Significant differences were determined by a *t*-test for two samples assuming unequal variances. *p* value ≤ 0.01 indicated a statistically significant difference. Underlined asterisks indicate statistical significance between males and females; asterisks indicate statistical significance between control and PLX5622-fed rats; vertical lines correspond to ± one standard deviation.

## Data Availability

Data is contained within the article and Appendix A.

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
