# Peer review of "Significant Sex Differences in the Efficacy of the CSF1R Inhibitor-PLX5622 on Rat Brain Microglia Elimination"

_pharmaceuticals, 2022, doi:10.3390/ph15050569_

Round 1

Reviewer 1 Report

The authors explored the effect of PLX5622 chow on microglia elimination in both female and male rats. The results provided essential information for the use of PLX5622 in microglia related studies among females and males rats. The present study was well designed and well written, which could be accepted in present form.

Author Response

Response: Thank you for the positive assessment of the submitted manuscript

Reviewer 2 Report

In this study, Aviv Sharon et al reported sex differences in the efficacy of the CSF1R inhibitor-PLX5622, given by diet, on the elimination of rat brain microglia . The authors suggest that sex differences must be taken into consideration when conducting studies with this CD115 inhibitor.

This brief report, is not easy to be read as the main text and the figure legends are not well separated.

Major comments.
The article never mentions the number of animals analyzed in each group; the material and methods section has no paragraph on statistical analysis and no statistical data is given in the legends ( in accordance with the scientific literature).

Figures 2 and 4 need to be improved. A title for the figure is lacking (not starting with a subtitle for panel A) as well as bars of significance between the different histograms which are compared. The number of animals in each group should be indicated as well as the statistical test used.

The mode of presentation of the results (mean +/- SD or mean +/- SEM) is not indicated.

For figures that show images, it is not specified whether the images were acquired from different brain areas of the same animal or from different animals.

No text about the interpretation of the results should be included in the figure legends or in the discussion ( to avoid repetition).

The interpretation of the results depends on the objective analysis of CD115 expression by microglial cells between male and female rats. Therefore, the authors should consider including analysis of CSF-1R expression in this study.

In the discussion, how do the suggested hypotheses relate to the gender differences in microglia density/morphology?

Minor comments

There are some grammatical and typographical errors, some of which are embarrassing (e.g.: anti-Iba-1 and anti-Ibl-1 lanes 106 and 107!!).

Reviewer 3 Report

The authors here attempt to describe the differences between male and female rats when the CSF1R inhibitor-PLX5622 is used. This brief report has scientific soundness and a high quality of presentation; however, it would be good if the authors could answer these questions before publication:

Since animals are administered ad libitum, how do the authors have control over the dose administered on each animal?
The authors did mention that the effect of PLX5622 is dose-dependent. Since male rats tend to weight more than female ones, could the male rats need a greater dose to observe the desired effect? Authors might expose their hypotheses in the discussion.

Lastly, authors must check the indentation of the figure 1 paragraph (lines 175, 176 and 177)

Author Response

Comment: Since animals are administered ad libitum, how do the authors have control over the dose administered on each animal?

Response: Thank you, In general, among the many STANDARD ways to administer drugs to laboratory animals, integration of the drugs with the chow is one of the preferable technologies.  ad libitum feeding of  chow that contain known drug concentration  IS A STANDARD PROTOCOL. The advantageous and disadvantageous of this technique in comparison to others including for example gastric gavage, intravenous, intraperitoneal or intracranial injections has been an intense subject of research, discussion and publications. Specifically more than 95% of the literature using CSF1R inhibitor (OVER 150 PUBLICATIONS IN THE LAST 5 YEARS) selected (for professional reasons) to administer the CSF1R inhibitors in in chow, given ad libitum. To maintain RELEVANCE vis a vis the past and future studies we selected to use the standard administration of PLX5622.

Comment: The authors did mention that the effect of PLX5622 is dose-dependent. Since male rats tend to weight more than female ones, could the male rats need a greater dose to observe the desired effect? Authors might expose their hypotheses in the discussion.

Response: We have used the "standard" protocol of ad libitum feeding of PLX5622 1200mg/Kg chow prepared by Research Diets Inc. In the last paragraph (253 -272) of the discussion we provided a tentative list of mechanisms that could account for the reported differences. Theoretically it is conceivable that larger doses of PLX5622 in the food might overcome differences. Nonetheless, the information presented in the manuscript shed objective light on how to interpret and deal with the growing use of CSF1R inhibitors in biomedical studies.

Comment: Lastly, authors must check the indentation of the figure 1 paragraph (lines 175, 176 and 177)

Response:  Thank you, We checked for the problem but could not find an indentation in the format version of the journal.

Round 2

Reviewer 2 Report

Dear authors

Thank you for the careful correction of the mansucript.
With the exception of the question concerning the analysis of CSF1-R expression, which I think is essential for the interpretation of the data, and which can only be approached through the prism of the effect of the chemical inhibitor, I would just like to emphasize that the statistical information should be mentioned in the text and main figures and not only in the supplemental figures/data.

I would suggest to add 2-3 sentences in the discusssion on the limits of the study regarding the absence of data on the expression of CSF1R in male versus female.

Sincerely

Author Response

Thank you,

We revised the manuscript as requested: (1) the statistical information is now (as requested) mentioned in the text and the figure legends. (2) As requested we added to the discussion a brief comment on the issue of CSF1R expression/functionality. 
